# Artificial Neural Network Modeling and Genetic Algorithm Multiobjective Optimization of Process of Drying-Assisted Walnut Breaking

**DOI:** 10.3390/foods12091897

**Published:** 2023-05-05

**Authors:** Taoqing Yang, Xia Zheng, Sriram K. Vidyarthi, Hongwei Xiao, Xuedong Yao, Yican Li, Yongzhen Zang, Jikai Zhang

**Affiliations:** 1College of Mechanical and Electrical Engineering, Shihezi University, Shihezi 832003, China; 20212009002@stu.shzu.edu.cn (T.Y.);; 2Key Laboratory of Northwest Agricultural Equipment, Ministry of Agriculture and Rural Affairs, Shihezi 832003, China; 3Key Laboratory of Modern Agricultural Machinery Corps, Shihezi 832003, China; 4Department of Biological and Agricultural Engineering, University of California, One Shields Avenue, Davis, CA 95616, USA; 5College of Engineering, China Agricultural University, Beijing 100080, China

**Keywords:** walnut, shell breaking, drying, artificial neural network, genetic algorithm, multi-objective optimization

## Abstract

This study combined an artificial neural network (ANN) with a genetic algorithm (GA) to obtain the model and optimal process parameters of drying-assisted walnut breaking. Walnuts were dried at different IR temperatures (40 °C, 45 °C, 50 °C, and 55 °C) and air velocities (1, 2, 3, and 4 m/s) to different moisture contents (10%, 15%, 20%, and 25%) by using air-impingement technology. Subsequently, the dried walnuts were broken in different loading directions (sutural, longitudinal, and vertical). The drying time (DT), specific energy consumption (SEC), high kernel rate (HR), whole kernel rate (WR), and shell-breaking rate (SR) were determined as response variables. An ANN optimized by a GA was applied to simulate the influence of IR temperature, air velocity, moisture content, and loading direction on the five response variables, from which the objective functions of DT, SEC, HR, WR, and SR were developed. A GA was applied for the simultaneous maximization of HR, WR, and SR and minimization of DT and SEC to determine the optimized process parameters. The ANN model had a satisfactory prediction ability, with the coefficients of determination of 0.996, 0.998, 0.990, 0.991, and 0.993 for DT, SEC, HR, WR, and SR, respectively. The optimized process parameters were found to be 54.9 °C of IR temperature, 3.66 m/s of air velocity, 10.9% of moisture content, and vertical loading direction. The model combining an ANN and a GA was proven to be an effective method for predicting and optimizing the process parameters of walnut breaking. The predicted values under optimized process parameters fitted the experimental data well, with a low relative error value of 2.51–3.96%. This study can help improve the quality of walnut breaking, processing efficiency, and energy conservation. The ANN modeling and GA multiobjective optimization method developed in this study provide references for the process optimization of walnut and other similar commodities.

## 1. Introduction

Walnut is the second largest nut in the world and has high edible and medicinal value. Breaking the shell to take the kernel is the premise of deep walnut processing. The shell is hard, and the gap between the shell and the kernel is small, leading to difficulty in breaking the shell to procure the kernel. Therefore, new walnut breaking processes must be explored.

The moisture content of walnuts has a considerable effect on their shell breaking characteristics [1]. At present, the research on the walnut drying process focuses on the drying kinetics and physicochemical qualities of walnuts [2,3,4,5], but few studies have linked the drying process with walnut breaking. In addition, adopting the proper loading direction can also improve the processing efficiency and quality of walnut breaking [6]. If drying is used as a pretreatment to assist in walnut breaking, a new walnut breaking process can be obtained through the selection of an appropriate loading direction. In addition, drying technology can ensure the food security of walnuts by reducing their moisture content [7].

Air-impingement technology ejects pressurized hot air through a nozzle and removes water through the impact and heating of hot air on a material. It is characterized by a high drying rate and high heat transfer coefficient. The application of this technology in walnut drying can increase the drying rate and reduce energy consumption.

The primary goal of walnut breaking is shell breaking and protecting the kernel. The shell-breaking rate is used to evaluate the difficulty of shell breaking, and it is the ratio of the number of cracked walnuts to the total number of walnuts in walnut breaking. High kernel rate is the ratio of the weight of the walnut kernel that is not completely crushed to the total weight of the walnut kernel. Whole kernel rate is the ratio of the weight of the whole walnut kernel to the total weight of the walnut kernel. High kernel and whole kernel rates are used to evaluate the integrity of the walnut kernel. At a drying temperature of 43 °C, walnut drying in an industrial hot air dryer requires a long drying time (more than 24 h) and consumes a large amount of energy (1284.6 MJ for natural gas and 85.1 MJ for electricity consumption per ton of dried walnuts) [8]. Therefore, the drying time and energy consumption must be minimised whilst ensuring the quality of walnut breaking. The optimization of the walnut breaking process depends on several parameters, including the shell-breaking rate (SR), high kernel rate (HR), whole kernel rate (WR), drying time (DT) and specific energy consumption (SEC).

The relationship between the process parameters of drying-assisted walnut breaking (including air velocity, temperature, moisture content, and loading direction) and the process evaluation index is highly nonlinear. Researchers have developed mathematical models to describe nonlinear models, such as theoretical, semitheoretical and empirical models. However, these models can accurately predict experimental data only under highly specific conditions, and no general equations are available to describe complete models [9]. Artificial intelligence technologies can model nonlinear systems where the relationships between variables are unknown [10]. An artificial neural network (ANN) is a kind of artificial intelligence technology that can simulate the behaviour of highly nonlinear systems dynamically [11]. ANNs have been successfully applied to the modeling and prediction of engineering problems, especially in areas where mathematical modeling methods fail [12]. ANN models have also been used in the moisture content prediction of pumpkin slice drying [13], prediction of the moisture ratio and color parameters of ginkgo biloba seed drying [14], prediction of the energy and exergy of mushroom slice drying [15] and moisture content prediction of carrot drying [16]. However, no study has examined its application in walnut drying and breaking. The model formula of an ANN is expressed by weights and thresholds. The essence of the ANN training process is the optimization of weights and thresholds. The initial weights and thresholds of the ANN are generated randomly. When the initial weights and thresholds are unreasonable, the convergence speed of the neural network is slow, and the training result is a local optimal value rather than a global optimal value. The genetic algorithm (GA) first preliminarily optimizes the initial weights and thresholds of the ANN, and then the neural network uses the optimized weights and thresholds for training to accelerate the convergence speed of the network and obtain the global optimal value.

The optimization of the walnut breaking process is multiobjective optimization (MOO), which is a method of finding the optimal target within the range of constraints based on the model. A GA is an optimization technique used to obtain the optimal value of a complex objective function by simulating biological evolutionary processes. Winiczenko, R applies a GA to the MOO during the apple cube drying process [17]. Raj, GVSB. used a GA to optimize the microwave vacuum drying process of dragon fruit slices [18]. MOO problems involve multiple objective functions, and these functions restrict one another. Therefore, obtaining a global optimal solution is difficult. Usually, MOO problems tend to obtain a set of optimal solutions called Pareto optimal solutions. A Pareto optimal solution is not dominated by other solutions in the solution space, and the improvement of one of the objectives requires the sacrifice of others [17]. The set of Pareto optimal solutions is called the Pareto front. The major goal of MOO is to find the Pareto front [19]. Obtaining the global optimal solution in the Pareto front in accordance with the specific optimization purpose is easy [20].

ANN modeling can be combined with GA MOO, which provides objective functions for MOO, and the genetic algorithm finds the optimal solution under the given constraint. This model is an effective method for predicting and optimizing any complex process parameters [21]. It has been applied in other fields [22,23].

In this study, drying was used as a pretreatment for walnut breaking for the first time, and the quality of walnut breaking and energy efficiency were utilised to evaluate the shell breaking process. An ANN and GA were applied to the modeling and optimization of the walnut breaking process for the first time. The objectives of this study are to: (1) analyse the effects of IR temperature (T), air velocity (V), moisture content (MC) and loading direction (D) on DT, SEC, HR, WR and SR; (2) develop an ANN nonlinear model with T, V, MC, and D as input variables, and DT, SEC, HR, WR, and SR as output variables; and (3) optimize the process parameters of drying-assisted walnut breaking by using a GA and minimize DT and SEC whilst improving the quality of walnut breaking. This study will provides an optimal process for drying-assisted walnut breaking.

## 2. Materials and Methods

### 2.1. Materials

Walnuts (cultivar Xinjiang Wen-185) of uniform size produced in Aksu, Xinjiang were selected and stored in a freezer (temperature: 5 °C) for use. The average weight, diameter, initial moisture content, and thickness of the walnuts were 16.52 ± 0.2 g, 36.43 ± 0.1 mm, 30.12 ± 0.65%, and 1.1 ± 0.1 mm, respectively.

### 2.2. Processing Equipment

Drying experiments were conducted using an air-impingement technology (Taizhou Senttech Infrared Technology Co., Ltd., Taizhou, China; temperature accuracy of ±0.1 °C, and power range of 0–2 kW). A principal diagram of the technology is shown in Figure 1. The air velocity at the nozzle was measured using an anemometer (AT816, SMART SENSOR, Thincol, Guangzhou, China; accuracy of ±0.1 m/s). Prior to the experiment, the technology was run for 30 min to stabilize the equipment. The walnut drying characteristics were used to predict when the walnuts reached the target moisture content. An electronic balance (BSM-5200.2, Shanghai Zhuojing Electronic Technology Co., Ltd., Shanghai, China; accuracy of 0.01 g) was adopted to weigh the walnuts and judge whether the walnuts reached the target moisture content. Drying was stopped when the walnuts reached the target moisture content, and continued otherwise. The drying experiments were performed at IR temperatures of 40 °C, 45 °C, 50 °C, and 55 °C; air velocities of 1, 2, 3, and 4 m/s, and moisture contents of 10%, 15%, 20%, and 25%. The dried walnuts were broken in three loading directions: sutural, longitudinal, and vertical.

### 2.3. Drying Experiments

The drying process parameters included IR temperature, air velocity and moisture content. IR temperature refers to the temperature of the hot air ejected from the nozzle, and air velocity refers to the velocity of the hot air ejected from the nozzle. The inner diameter of the nozzle was 10 mm, and the number of nozzles was 18. The distribution of the nozzles is shown in Figure 1. Moisture content refers to the moisture content (dry basis) of the walnut at the end of drying.

The IR temperature range was set to be 40–55 °C to ensure drying efficiency and product quality [5]. The maximum air velocity of the equipment was 4 m/s. To sufficiently study the influence of air velocity on walnut breaking, the range of air velocity was set to 1–4 m/s. To prevent production reduction caused by walnuts with a moisture content below the safe storage moisture content of 8%, the minimum moisture content of the walnuts was set to 10%. The range of moisture content was 10–25% [1]. The drying experiments were conducted at IR temperatures of 40 °C, 45 °C, 50 °C, and 55 °C; air velocities of 1, 2, 3, and 4 m/s, and moisture contents of 10%, 15%, 20%, and 25%. The dried walnuts were broken in three loading directions: sutural, longitudinal and vertical. A total of 192 experimental groups (4 × 4 × 4 × 3) were established.

### 2.4. Loading Direction(L)

Breaking experiments were performed using a DF-9000 dynamic and static universal material testing machine, and the loading rate was set to 10 mm/min. The walnuts’ loading direction is shown in Figure 2.

### 2.5. High Kernel Rate (HR), Whole Kernel Rate (WR) and Shell-Breaking Rate (SR)

The integrity of walnut kernels was evaluated with the high kernel and whole kernel rates. The walnut kernels were divided into Classes A, B, C, and D according to their integrity. Whole walnut kernels belonged to Class A, half walnut kernels belonged to Class B, quarter walnut kernels were classified as Class C and the rest of the crushed walnut kernels belonged to Class D.

High kernel rate was calculated using Equation (1), and whole kernel rate was calculated with Equation (2).
(1)$$HR=\frac{M_{A}+M_{B}+M_{C}}{M_{A}+M_{B}+M_{C}+M_{D}}\times 100\%$$
(2)$$WR=\frac{M_{A}}{M_{A}+M_{B}+M_{C}+M_{D}}\times 100\%$$

The difficulty of shell breaking was evaluated by the shell-breaking rate (*SR*). *SR* is the ratio of the number of walnuts that are cracked by more than 3/4 to the total number of walnuts in the experiment. SR was calculated with Equation (3).
(3)$$SR=\frac{L}{N}\times 100\%$$
where *L* is the number of walnuts that are cracked by more than 3/4; *N* is the total number of walnuts, *M_A_* is the weight of Class A walnut kernels, *M_B_* is the weight of class B walnut kernels; *M_C_* is the weight of Class C walnut kernels and *M_D_* is the weight of Class D walnut kernels.

### 2.6. Drying Time and Specific Energy Consumption

The walnuts were dried in a hot-air dryer at 105 °C for 24 h to measure the dry weight, *W_d_* [24]. The moisture content of the walnuts at time T was calculated by Equation (4) [25]. The time required for the walnuts to dry to the target moisture content was the drying time.
(4)$$M_{t}=\frac{W_{T}-W_{d}}{W_{d}}\times 100\%$$

The energy required to dry 1 kg of walnuts was defined as the specific energy consumption, which was calculated with Equation (5).
(5)$$SEC=\frac{E}{m_{walnut}}$$
where *E* is the electrical energy consumed during drying and *m_walnut_* is the total walnut mass during drying.

### 2.7. Shell Kernel Clearance (SKC)

Shell kernel clearance refers to the distance between the walnut shell and the walnut kernel, and it was calculated using Equation (6) [26].
(6)$$SKC=\frac{A+B+C-a-b-c-6d}{6}$$
where *A* is the maximum longitudinal dimension of the inner wall of the walnut shell (mm), *B* is the maximum vertical dimension of the inner wall of the walnut shell, (mm), *C* is the maximum sutural dimension of the inner wall of the walnut shell (mm), *a* is the maximum longitudinal dimension of the walnut kernel (mm), *b* is the maximum vertical dimension of walnut kernel, (mm), *c* is the maximum sutural dimension of the walnut kernel (mm) and *d* is the thickness of the walnut shell (mm).

### 2.8. Walnut Shell Hardness

Hardness is one of the important factors that affect walnut breaking. A DF-9000 dynamic and static universal material testing machine was used for the walnut breaking experiment, and the loading rate was set to 10 mm/min. Hardness H was calculated using Equation (7) [27].
(7)$$H=\frac{F_{r}}{D_{r}}$$
where *F_r_* is the maximum breaking force and *D_r_* is the deformation when the maximum breaking force is reached. *D_r_* can be obtained from the DF-9000 dynamic and static universal material testing machine.

### 2.9. Artificial Neural Network (ANN) Model

Back propagation ANN (BP-ANN) is a feed-forward network-trained model in accordance with the error-back propagation algorithm, and it is one of the most widely used ANN models. The model of the process of drying-assisted walnut breaking was established using BP-ANN. The topology of the ANN model consisted of three layers (input layer, hidden layer and output layer) and two transfer functions between the three layers, as shown in Figure 3. The input layer had four factors: IR temperature (T), air velocity (V), moisture content (MC), and loading direction (D). The output layer had five factors: drying time (DT), specific energy consumption (SEC), high kernel rate (HR), whole kernel rate (WR), and shell-breaking rate (SR). The range of the number of neurons in the hidden layer was determined with the Equations (8)–(11). The optimal number of neurons in the hidden layer was obtained by trial and error. Nonlinear transfer functions, including tansig sigmoidal and logsig sigmoidal functions, were used in the input and hidden layers, and a linear transfer function (Pureline) was employed in the hidden layers and output layers. The topology of neural networks influences two of the most important evaluation criteria of neural network training: generalization and training time [28]. Improper topology of the neural network causes many redundancies, which makes the neural network fall into local optimization and considerably prolongs the training time [29]. Therefore, determining the optimal neural network topology is important.

The range of the number of neurons in the hidden layer was determined using the following equations [30]:(8)j<n−1
(9)$$j<\sqrt{m+n}+a$$
(10)$$j=\log_{2}n$$
(11)j=2n+1
where *j* is the number of neurons in the hidden layer, *n* is the number of factors in the input layer, *m* is the number of factors in the output layer and *a* is a constant between 0 and 10.

The union of Equations (8)–(11) was determined as the range of the number of neurons in the hidden layer to ensure that the optimal number of neurons in the hidden layer could be achieved. Generally, the number of neurons in the hidden layer should be more than one; then, the range is as follows: 1 < *j* < 13. To obtain the variation in neural network performance beyond the range, the range of the number of neurons in the hidden layer was determined as 1 < *j* ≤ 13 in this study.

The coefficient of determination (*R^2^*) and root mean square error (*RMSE*) were calculated to determine the optimal number of neurons in the hidden layer and the transfer function in the input and hidden layers by using Equations (12) and (13), respectively [31,32].
(12)$$RMSE=\sqrt{\frac{\sum_{i=1}^{N}{\left(y_{act,i}-y_{pre,i}\right)}^{2}}{N}}$$
where *y_act_*_,*i*_ is the actual value and *y_pre_*_,*i*_ is the predicted value of the sample. In general, the smaller the *RMSE* value is, the higher the model accuracy is.
(13)$$R^{2}=1-\frac{\sum_{i=0}^{N}{\left(y_{i}-\hat{y_{i}}\right)}^{2}}{\sum_{i=0}^{N}{\left(y_{i}-\bar{y_{i}}\right)}^{2}}$$
where *y_i_* is the actual value of the sample, $\hat{y}$ is the predicted value of the sample, $\bar{y}$ is the average of the actual values, and *N* is the number of test samples. The larger the *R^2^* value is, the better the predictive ability of the model is.

After determining the optimal topology structure of the neural network, the neural network was trained. A total of 192 groups of experimental data (4 × 4 × 4 × 3) were established. One group consisted of four input variables and five output variables. To ensure the effectiveness of neural network training and the accuracy of the test data, 80% of the experimental data (154 groups) were randomly selected as training data, and 20% (38 groups) were adopted as test data [33]. The algorithm flow of the neural network is shown in the Figure 4.

The weights and thresholds of the neural network were obtained after the training was completed, and the neural network model was established. The neural network model had four parts: the input variables were normalized; the weight, threshold, and transfer function of the hidden layer were inputted into Equations (15) and (16) to obtain the output of the hidden layer; the weight, threshold, and transfer function of the output layer were inputted into Equation (17) to obtain the output of the output layer and the output of the output layer was reversely normalized to obtain the output variables.

The input variables were normalized, and the normalized range was (−1, 1).
(14)$$x_{i}^{g}=\frac{2(x_{i}-x_{i,\min})}{x_{i,\max}-x_{i,\min}}-1$$
where $x_{i}^{g}$ (*i* = 1, 2, 3, 4) is the normalized input variable, *x_i_* is the input variable; *x*_1_ is T, *x*_2_ is V, *x*_3_ is MC, *x*_4_ is D, *x_i_*_,min_ is the minimum of the training data for the input variable *x_i_* and *x_i_*_,max_ is the maximum of the training data for input variable *x_i_*.

The output in the hidden layer is:(15)$$h_{j}=\log\mathrm{sig}(\sum w_{ij}^{h}x_{i}^{g}+b_{j}^{h})$$
(16)$$\log\mathrm{sig}(x)=\frac{1}{1+\exp(-x)}$$
where *h_j_* (*j* = 1, 2, 3, 4, 5, 6, 7, 8, 9, 10, 11, 12) refers to the outputs of each neuron in the hidden layer, $w_{ij}^{h}$ is the weight between the input layer and the hidden layer, and $b_{j}^{h}$ is the threshold in the hidden layer. The values of weights $w_{ij}^{h}$ and threshold $b_{j}^{h}$ are used in Equation (15).

The output in the output layer is:(17)$$y_{k}^{g}=purelin(\sum w_{jk}^{o}h_{j}+b_{k}^{o})=\sum w_{jk}^{o}h_{j}+b_{k}^{o}$$
where $y_{k}^{g}$ (*k* = 1, 2, 3, 4, and 5) refers to the outputs of each neuron in the output layer, $w_{jk}^{o}$ is the weight between the hidden layer and the output layer, and $b_{k}^{o}$ is the threshold in the output layer. The values of weights $w_{jk}^{o}$ and threshold $b_{k}^{o}$ are used in Equation (17).

Reverse normalization of the output in the layer output was implemented as follows:(18)$$DT=\frac{\left(y_{1,\max}-y_{1,\min})(y_{1}^{g}+1\right)}{2}+y_{1,\min}$$
(19)$$SEC=\frac{\left(y_{2,\max}-y_{2,\min})(y_{2}^{g}+1\right)}{2}+y_{2,\min}$$
(20)$$HR=\frac{\left(y_{3,\max}-y_{3,\min})(y_{3}^{g}+1\right)}{2}+y_{3,\min}$$
(21)$$WR=\frac{\left(y_{4,\max}-y_{4,\min})(y_{4}^{g}+1\right)}{2}+y_{4,\min}$$
(22)$$SR=\frac{\left(y_{5,\max}-y_{5,\min})(y_{5}^{g}+1\right)}{2}+y_{5,\min}$$
where *y_k_*_,min_ is the minimum of the training data for output variable *y_k_* and *y_k,_*_max_ is the maximum of the training data for the output variable *y_k_*.

### 2.10. Optimization of Artificial Neural Network Using Genetic Algorithm

The neural network is sensitive to the initial weights and thresholds. Therefore, the training results are greatly affected by the initial weights and thresholds, and they easily fall into the local minimum [34]. GA is a global optimization method based on the principle of biological evolution, namely, survival of the fittest [35]. By optimizing the initial weights and thresholds, GA can move the neural network training process from the local optimal domain to the global optimal domain [36]. Therefore, for the walnut breaking process model, the initial weights and thresholds of the ANN can be optimized by the GA first, followed by ANN training with the optimized initial weights and thresholds, resulting in global optimal weights and thresholds. The parameters of the GA are shown in Table 1.

The GA assisted by ANN modeling was divided Into three parts. First, by analyzing R^2^ and RMSE, the optimal topology of the neural network was selected, and then the structure of the initial weights and thresholds was obtained. Second, the weights and thresholds of the neural network were adopted as the optimization object, the error of the neural network model was used as the fitness function, and the initial weights and thresholds were optimized using the GA. Lastly, the optimized weights and thresholds were used for neural network training, and the global optimal weights and thresholds were obtained. The algorithm flow is shown in Figure 5.

### 2.11. Multiobjective Optimization

The MOO of the walnut breaking process was applied to reduce the energy consumption and drying time while improving the quality of walnut breaking. The optimization goal was to minimize DT and SEC and maximize HR, WR, and SR. The upper and lower limits of the input variables were the upper and lower limits of the experimental conditions in the walnut breaking experiment, respectively, as shown in Equation (23). In terms of the loading direction, 1 was the sutural direction, 2 was the Longitudinal direction, and 3 was the Vertical direction.

Five objective functions and four constraints were entered in MATLAB, and the GAMULTIOBJ function was used to obtain the Pareto optimal solution. The parameters of the GAMULTIOBJ function are listed in Table 2.
(23)$$objectives=\left\{\begin{array}{c} Min\;DT\left(T,V,MC,D\right)\\ Min\;SEC\left(T,V,MC,D\right)\\ Max\;HR\left(T,V,MC,D\right)\\ Max\;WR\left(T,V,MC,D\right)\\ Max\;SR\left(T,V,MC,D\right)\\ 40\leq T\leq 55\;{}^{\circ}C\\ 1\leq V\leq 4m\cdot x^{-1}\\ 10\leq MC\leq 25\%\\ D=\left(1,2,3\right) \end{array}\right.$$

### 2.12. Statistical Analysis

The experimental data were processed with Microsoft Excel 2019, and drawing was performed with Origin 2019b. ANN modeling and GA MOO were implemented in MATLAB R2021a.

## 3. Results and Discussion

### 3.1. Drying Time and Specific Energy Consumption

The effect of IR temperature on DT and SEC under a constant air velocity of 3 m/s and moisture content of 15% is shown in Figure 6a. DT was greatly influenced (*p* < 0.01) by the IR temperature. DT decreased with increasing IR temperature due to increments in heat and moisture transfer rates and thermal radiation intensity [37]. With increasing IR temperature, DT decreased considerably, but SEC did not decrease rapidly. The increase in IR temperature led to an increase in equipment power, and thus increased the equipment’s energy consumption [38]. Figure 6b shows the effect of air velocity on DT and SEC under a constant IR temperature of 50 °C and moisture content of 15%. DT decreased considerably with increasing air velocity due to the accelerated discharge of water vapor from the drying chamber caused by an increase in the heat convection rate [39]. SEC decreased considerably with increased air velocity. First, the increase in air velocity reduced the drying time, which refers to the time for the equipment to operate. Second, fan power consumed a small portion of the total power of the equipment, and the increase in air velocity did not greatly improve the total power of the equipment. Thus, the energy consumption did not increase much. As shown in Figure 6c, DT and SEC decreased greatly with increasing moisture content under an IR temperature of 50 °C and air velocity of 3 m/s. Notably, the DT and SEC of the sample with 15% moisture content decreased by 45.11% and 47.79%, respectively, compared with that of the sample with 10% moisture content. This result was obtained because the drying rate decreased substantially at the last stage of drying, resulting in a large increase in drying time [40]. The energy consumption of equipment was directly related to the length of drying time, so SEC decreased rapidly with increasing moisture content [20]. Among the three drying parameters, moisture content had the greatest influence on drying time and SEC. Thus, increasing the moisture content is the best approach for minimizing drying time and SEC, provided that the quality of walnut breaking is not reduced.

### 3.2. High Kernel Rate and Whole Kernel Rate

As shown in Figure 7a, at a constant air velocity of 3 m/s, moisture content of 15% and longitudinal loading direction, IR temperature had no notable influence on the high kernel and whole kernel rates. No remarkable difference in shell kernel clearance was observed among the four groups. The effect of air velocity on the high kernel rate was not substantial at a constant IR temperature of 50 °C, moisture content of 15% and in the longitudinal loading direction, as shown in Figure 7b. With the increase in air velocity, the whole kernel rates of Groups 3 and 4 decreased compared with those of Groups 1 and 2. The whole kernel rate is related to shell kernel clearance, and it can only be guaranteed when the deformation of shell breaking is smaller than the shell kernel clearance [26]. Therefore, the decrease in whole kernel rate may be due to the decrease in shell kernel clearance. Figure 7c shows that the high kernel rate and whole kernel rate decreased significantly with an increase in the moisture content from 10% to 25% at a fixed IR temperature of 50 °C, an air velocity of 3 m/s, and longitudinal loading direction. Wang Jiannan et al. had the same finding [1]. As presented in Figure 7d, under a constant IR temperature of 50 °C, air velocity of 3 m/s and moisture content of 15%, the high kernel and whole kernel rates of sutural direction were the highest, followed by those of the longitudinal and vertical directions in turn. Under the same drying conditions, the shell kernel clearance of the three groups was almost undifferentiated. “Shell kernel clearance” refers to overall shell kernel clearance and is the mean value of shell kernel clearance in three directions of the walnut. However, the shell kernel clearance in different directions obviously differed, which resulted in a great difference in the high kernel and the whole kernel rates of different loading directions. In conclusion, moisture content had the greatest influence on the high kernel and whole kernel rates, and reducing the moisture content helped improve the high kernel and whole kernel rates. However, this result is contradictory to the minimization of drying time and SEC. The optimization of the walnut breaking process needs to achieve balance between energy efficiency and quality of walnut breaking.

### 3.3. Shell-Breaking Rate

Figure 8a shows the effects of different IR temperatures on the shell-breaking rate when the air velocity, moisture content and loading direction were 3 m/s, 15%, and longitudinal, respectively. The shell-breaking rate and hardness of Groups 3 and 4 increased compared with those of Groups 1 and 2 when the IR temperature continued to increase to 55 °C. The water loss rate of the walnut shells increased with the increasing IR temperature, which accelerated the change in porosity and rapidly increased the density of walnut shells, leading to an increase in walnut shell hardness [41]. With high walnut shell hardness, high breaking force would need be exerted under the same deformation, and walnut shells are more likely to be broken. Feizollah Shahbazi also believed that the shell-breaking rate was related to the hardness of walnut shell [42]. As shown in Figure 8b, at a constant IR temperature of 50 °C, moisture content of 15%, and longitudinal loading direction, air velocity had no notable influence on the shell-breaking rate. No remarkable difference in the hardness was observed among the four groups of experiments. As shown in Figure 8c, at a constant IR temperature of 50 °C, air velocity of 3 m/s, and longitudinal loading direction, the shell-breaking rate decreased significantly with increasing moisture content. The increase in walnut moisture content reduced the walnut shell hardness, indicating a small shell-breaking force was required to achieve the same deformation. Feizollah Shahbazi found that the decrease in the shell-breaking rate is due to the softening of walnuts at higher moisture contents [42]. At the same time, as the moisture content of walnuts decreases, the hygrothermal stress in the drying process causes the shell to produce microcracks and other damage, and the shell becomes prone to fracture and breakage [43]. Figure 8d show that the shell-breaking rate was the highest in the vertical direction when the IR temperature was fixed at 50 °C, the air velocity was 3 m/s, and the moisture content was 15%. The thickness and structure of the walnut shells in different directions varied, resulting in different shell-breaking rates in different loading directions. Liu Kui et al. also found that walnuts are likely to be broken in the vertical direction [44]. In conclusion, the influence law of moisture content on the shell-breaking rate is similar to that of the high kernel and whole kernel rates. In addition, the shell-breaking rate can be optimized by selecting an appropriate IR temperature and loading direction.

### 3.4. Construction of Artificial Neural Network Model

The construction of the ANN model proceeded through three steps. The first step was to determine the optimal topology of the ANN. The second step was to optimize the initial weight and threshold of the ANN by using a GA. In the third step, the ANN used the optimized weights and thresholds for network training.

To obtain the optimal topology of the neural network, the transfer function (‘transig’ sigmoidal or ‘logsig’ sigmoidal) between the input layer and the hidden layer and the number of neurons in the hidden layer (2–13) needed to be determined. Nonoptimized weights and thresholds for network training can cause a large deviation in R^2^ and RMSE under the same topology structure. Therefore, the results of the same topology structure were adopted as the mean value of 10 simulations. The simulation results of different ANN topologies is shown in Table 3.

The highest R^2^ values of DT, SEC, HR, WR, and SR were achieved in Group 23 (‘logsig’ sigmoidal + 12 neurons). The minimum RMSEs of DT, SEC, HR, WR, and SR were achieved in Groups 21, 22, 23, 10 and 10, respectively. The subminimum RMSEs of DT, SEC and WR appeared in the Group 23, with a deviation of 10%, 2.25% and 2.93% from the minimum RMSE, respectively. The RMSE of SR in Group 23 differed by 7.51% from the minimum RMSE of SR. In summary, the topology of Group 23 (‘logsig’ sigmoidal + 12 neurons) was the optimal topology.

The results showed that different topologies had a great influence on R^2^ and RMSE. The ‘logsig’ sigmoidal function between the input layer and hidden layer was much better than the ‘transig’ sigmoidal function. The larger the number of neurons in the hidden layer was, the better the prediction ability of the model was. However, having too many neurons can lead to overfitting, resulting in reduced model accuracy. Therefore, the number of neurons in the hidden layer was set as 12 based on the optimal topology and GA to optimize the initial weights and thresholds of the neural network. The parameters of the GA are shown in Table 1.

The neural network was trained with the optimized weights and thresholds. A comparison of the results of the BP neural network model and the GA neural network model (GA-ANN) is shown in Table 4.

In the GA-ANN model, the R^2^ values of DT, SEC, HR, WR and SR were all above 0.990; RMSE decreased by 32.8%, 23.14%, 10.96%, 21.99%, and 50.5% compared with the BP model, respectively. The GA-ANN model had a remarkable optimization effect, indicating that the GA-ANN model removed the local optimal value and obtained the global optimal value.

The experimental data and predicted values of the GA-ANN model are shown in Figure 9. The GA-ANN model demonstrated a sufficient prediction ability and can be further applied to the MOO of the process of drying-assisted walnut breaking.

### 3.5. Multi-Objective Optimization

The Pareto optimal set, which included 30 optimal solutions, was obtained using the GAMULTIOBJ function, as shown in Table 5. The selection of specific optimal solutions depended on the primary purpose of the process. For example, Group 1 was selected as the optimal process for the highest HR of 89%. Group 3 was a slightly worse process because HR and SR decreased by 0.2% and 3.6%, respectively, and DT increased by 12.9%. Moreover, it had a similar SEC of 16.78 MJ/kg relative to Group 1, but WR increased by 0.6%. In addition, the lowest DT of 87.3 min (90.9% lower than that of Group 1) and the lowest SEC of 0.81 MJ/kg (95.2% lower than that of Group 1) were obtained in Group 30. Under this processing condition, HR was 46.1% (48.2% lower than that of Group 1), WR was 22.4% (56% lower than that of Group 1) and the SR was 61.8% (33.4% lower than that of Group 1). Given that the primary purpose of MOO was to ensure the quality of walnut breaking, Group 30 was not be selected as the optimal solution. Group 7 had the highest SR of 99.8% (7.5% higher than that of Group 1), HR of 81.7% that of (8.2% lower than that of Group 1), and WR of 43.7% (14.1% lower than that of Group 1). DT decreased by 39.0%, and SEC decreased by 23.3%, relative to Group 1. The reason was that Group 7 adopted the highest possible values of IR temperature and air velocity to reduce the drying time and energy consumption. In addition, the drying rate of walnuts at the last stage of drying was extremely low. Group 7 slightly increased the moisture content, which considerably reduced the drying time and energy consumption. Group 7 reduced the drying time and energy consumption as much as possible without notably reducing the quality of walnut breaking. In conclusion, Group 7’s parameters (T = 54.9, V = 3.66, MC = 10.9, vertical loading direction) were selected as the optimal parameters for the process of drying-assisted walnut breaking.

Verification experiments were conducted under the optimal parameters to verify the optimization results. As shown in Table 6, the errors between the predicted values and the experimental data were 3.56%, 3.88%, 2.51%, 3.32%, and 3.96% for DT, SEC, HR, WR, and SR, respectively, indicating good accuracy. The ANN model optimized by the GA could accurately predict the process parameters of drying-assisted walnut breaking, and the MOO results of the GA were effective.

## 4. Conclusions

A process of drying-assisted walnut breaking was developed in this study. An ANN model optimized by a GA (GA-ANN) was established to simulate the effects of input variables (T, V, MC, and D) on output variables (DT, SEC, HR, WR, and SR). The GA-ANN model demonstrated sufficient prediction ability, with coefficients of determination of 0.996, 0.998, 0.990, 0.991, and 0.993 for DT, SEC, HR, WR, and SR, respectively. RMSE decreased by 32.8%, 23.14%, 10.96%, 21.99%, and 50.5% compared with those in the ANN model. On the basis of the GA-ANN model, the GA was applied to MOO of the walnut breaking process with the aim of minimizing DT and SEC and maximizing HR, WR and SR. The optimal process parameters were determined to be T = 54.9, V = 3.66, MC = 10.9, and vertical loading direction, which led to DT = 585.1 min, SEC = 12.86 MJ/kg, HR = 81.7%, WR = 43.7%, and SR = 99.8%. The model that combines ANN and GA was proven to be effective in predicting and optimizing the process parameters of walnut breaking. The predicted values under the optimized process parameters fitted the experimental data well, with low relative error values of 3.56%, 3.88%, 2.51%, 3.32%, and 3.96% for DT, SEC, HR, WR, and SR, respectively. The process of drying-assisted walnut breaking considerably improved the quality of walnut breaking and reduced the energy consumption and drying time. The ANN modeling and the method of GA MOO developed in this study could be applied to other similar commodities.

## Figures and Tables

**Figure 1 foods-12-01897-f001:**
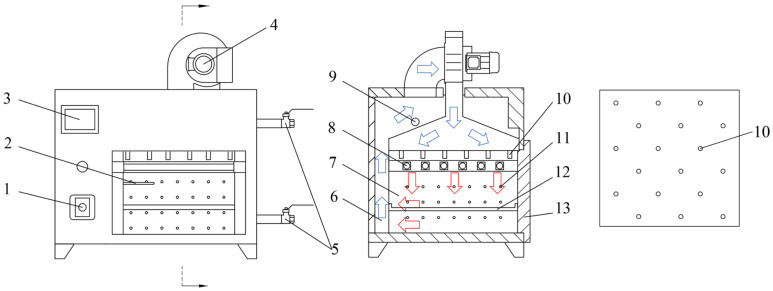
The principal diagram of the air-impingement technology. (1) air velocity adjustment knob; (2) Temperature transducer; (3) temperature control touch panel; (4) centrifugal fan; (5) wet discharge valve; (6) drying outer chamber; (7) drying inner chamber; (8) infrared heating tube; (9) weephole; (10) air nozzle; (11) air outlet; (12) drying tray; (13) door.

**Figure 2 foods-12-01897-f002:**
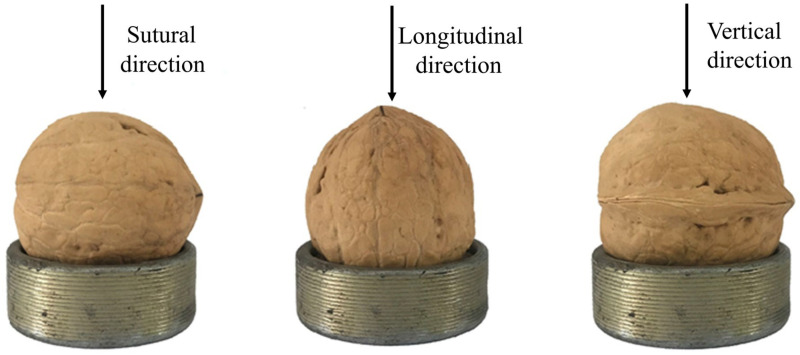
Walnuts’ loading direction.

**Figure 3 foods-12-01897-f003:**
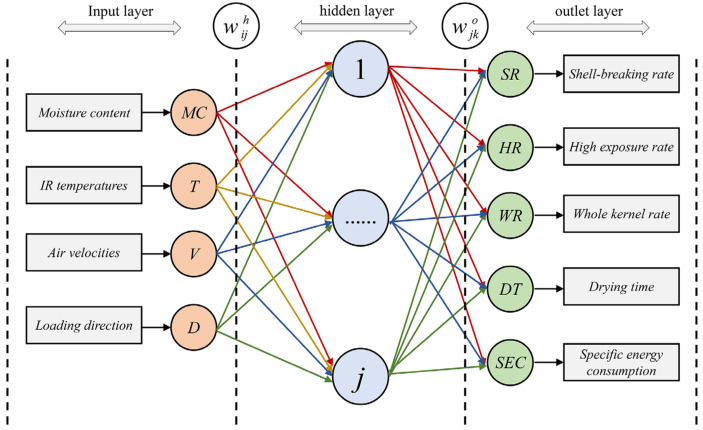
Topology of ANN model.

**Figure 4 foods-12-01897-f004:**
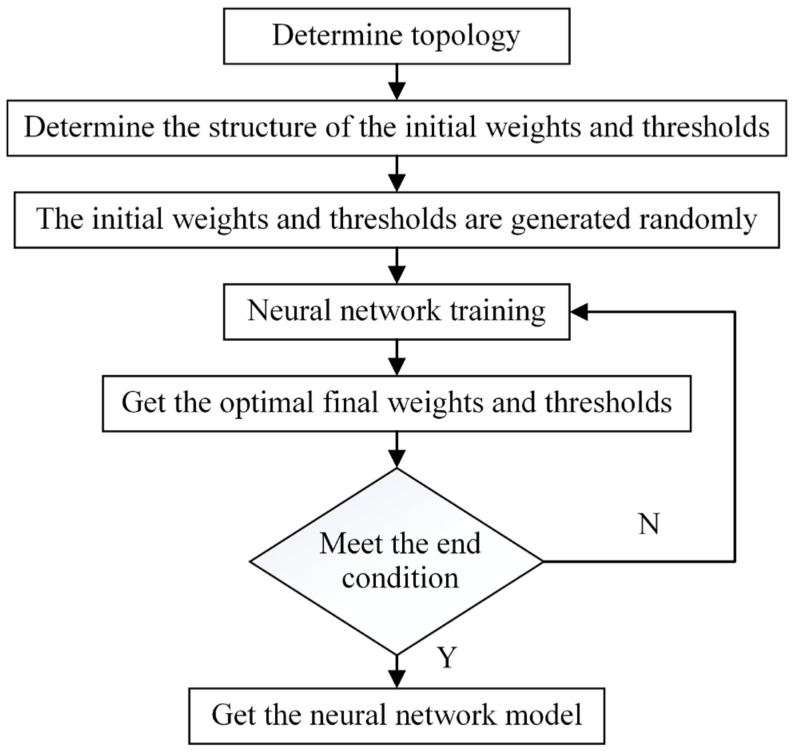
Flow of ANN algorithm.

**Figure 5 foods-12-01897-f005:**
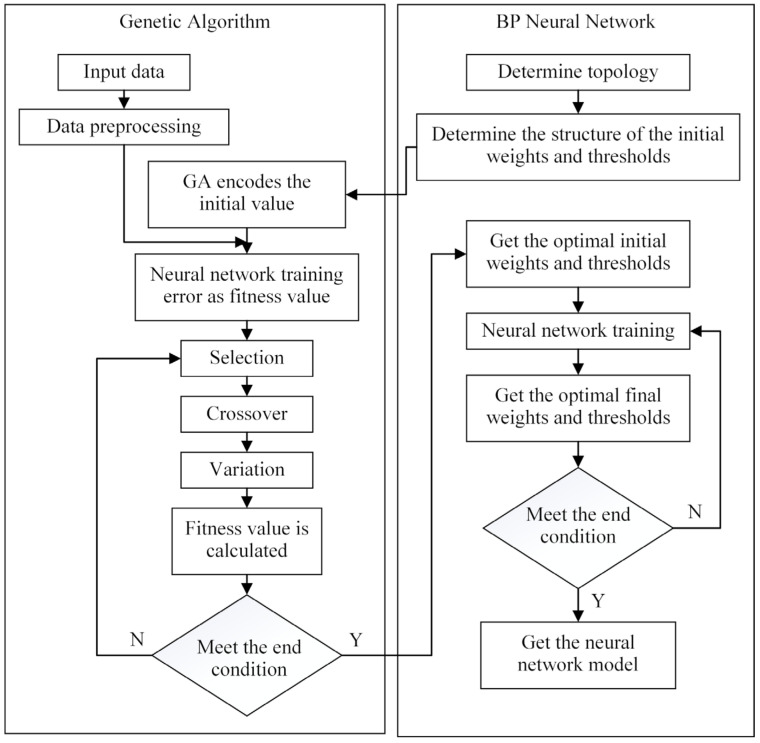
Flow of GA-ANN algorithm.

**Figure 6 foods-12-01897-f006:**
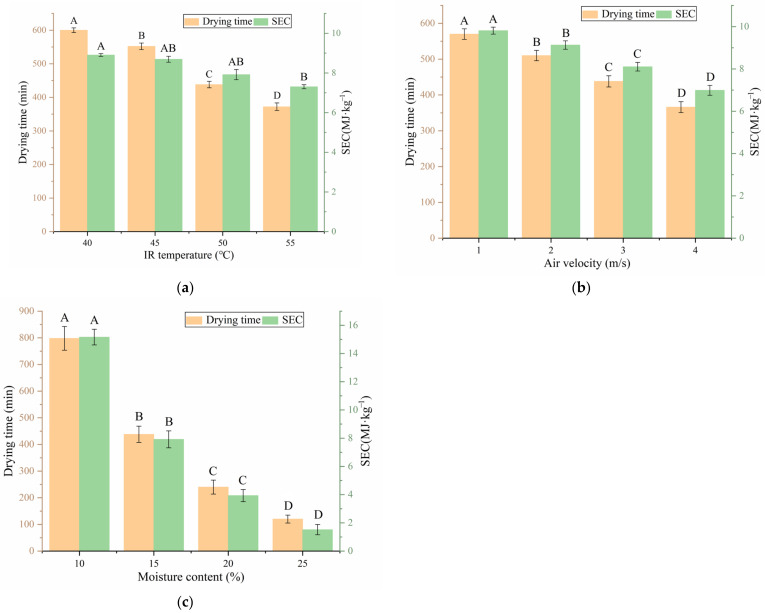
Effects of T, V and MC on DT and SEC. (**a**) Air velocity was 3 m/s and moisture content was 15%. (**b**) IR temperature was 50 °C and moisture content was 15%. (**c**) IR temperature was 50 °C and air velocity was 3 m/s. Note: different letters of columns of same color indicate significant differences between the mean values (*p* < 0.01).

**Figure 7 foods-12-01897-f007:**
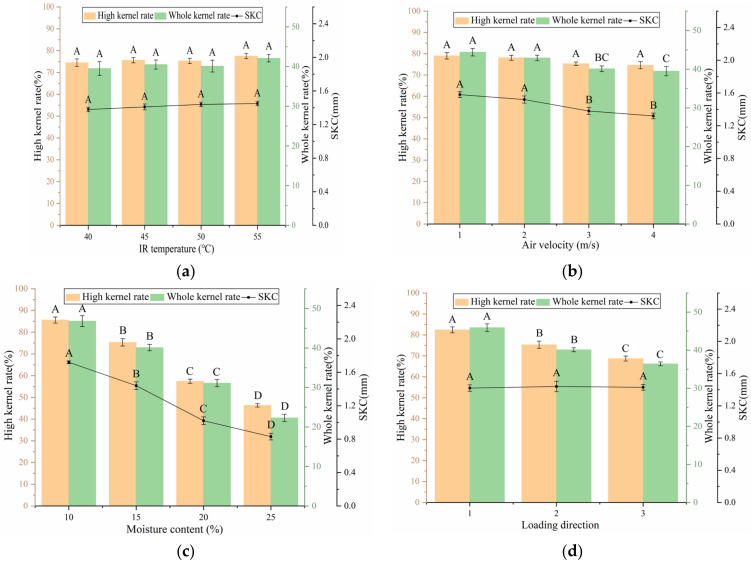
Effects of T, V, MC and DT on HR and WR. (**a**) Air velocity was 3 m/s, moisture content was 15% and loading direction was Longitudinal. (**b**) IR temperature was 50 °C, moisture content was 15% and loading direction was Longitudinal. (**c**) IR temperature was 50 °C, air velocity was 3 m/s and loading direction was Longitudinal. (**d**) IR temperature was 50 °C, air velocity was 3 m/s and moisture content was 15%. Note: different letters of columns of the same color indicate significant differences between the mean values (*p* < 0.01).

**Figure 8 foods-12-01897-f008:**
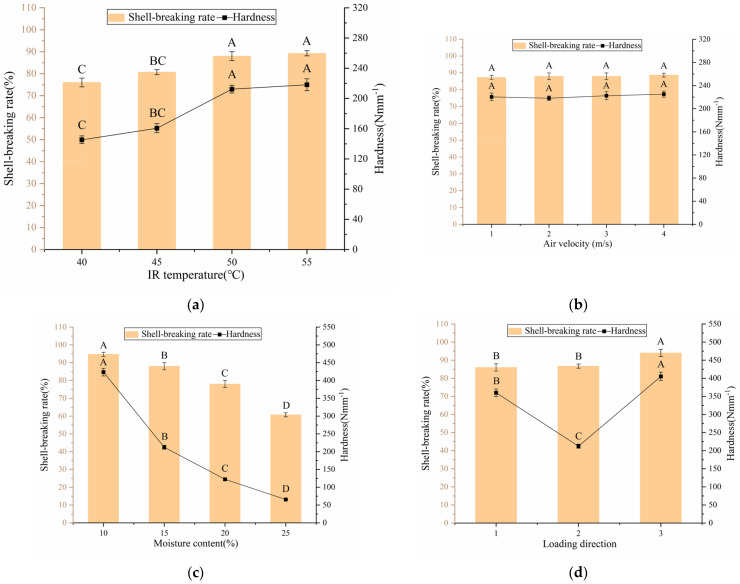
Effects of T, V, MC and DT on SR. (**a**) Air velocity was 3 m/s, moisture content was 15% and loading direction was Longitudinal. (**b**) IR temperature was 50 °C, moisture content was 15% and loading direction was Longitudinal. (**c**) IR temperature was 50 °C, air velocity was 3 m/s and loading direction was Longitudinal. (**d**) IR temperature was 50 °C, air velocity was 3 m/s and moisture content was 15%. Note: different letters of columns of the same color indicate significant differences between the mean values (*p* < 0.01).

**Figure 9 foods-12-01897-f009:**
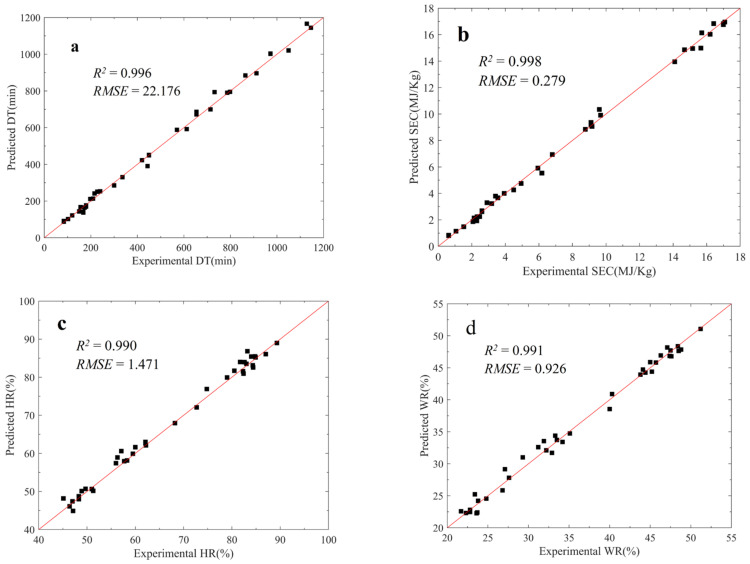
Experimental data and predicted values of the GA-ANN model: (**a**) drying time; (**b**) specific energy consumption; (**c**) high kernel rate; (**d**) whole kernel rate, and (**e**) shell-breaking rate.

**Table 1 foods-12-01897-t001:** Parameters of genetic algorithm.

Population Type	Double Vector
variable range	(−3, 3)
Population size	200
Number of generations	1000
Crossover rate	90%
Mutation rate	10%

**Table 2 foods-12-01897-t002:** The parameters of GAMULTIOBJ function.

Population Type	Double Vector
Pareto front population fraction	0.3
Population size	120
Number of generations	1000
Crossover rate	90%
Mutation rate	10%

**Table 3 foods-12-01897-t003:** Simulation results of different ANN topologies (shaded group is the optimal topology; bold font indicates the optimal solution).

Group	Transfer Function of the Hidden Layer	Number of Neurons in the Hidden Layer	DT		SEC		HR		WR		SR	
			*R* ^2^	*RMSE*	*R* ^2^	*RMSE*	*R* ^2^	*RMSE*	*R* ^2^	*RMSE*	*R* ^2^	*RMSE*
1	Tansig	2	0.6101	87	0.9170	0.7809	0.7262	3.6053	0.8348	2.4655	0.7695	3.4578
2		3	0.7330	70	0.9430	0.6930	0.8490	2.7626	0.9029	1.9311	0.8680	2.9473
3		4	0.8053	62	0.9290	0.5373	0.8953	2.2088	0.9284	1.6161	0.9034	2.6510
4		5	0.8455	50	0.9113	0.5288	0.9198	2.2477	0.9377	1.5535	0.9231	2.5961
5		6	0.8688	42	0.9172	0.4820	0.9331	2.0987	0.9411	1.3893	0.9231	2.8564
6		7	0.8858	38	0.9234	0.4868	0.9430	1.9751	0.9474	1.3514	0.9245	2.5263
7		8	0.8965	36	0.9204	0.4876	0.9509	1.9454	0.9522	1.3170	0.9246	2.6684
8		9	0.9030	37	0.9261	0.4557	0.9561	1.8920	0.9560	1.2482	0.9214	2.3708
9		10	0.9113	36	0.9256	0.4839	0.9589	1.9481	0.9556	1.3308	0.9207	2.4248
10		11	0.9176	34	0.9276	0.4794	0.9619	1.7221	0.9570	**1.1531**	0.9158	**2.3483**
11		12	0.9207	37	0.9296	0.4642	0.9633	1.7937	0.9573	1.2995	0.898	2.4721
12		13	0.9018	38	0.9248	0.4782	0.9610	1.8238	0.9568	1.3210	0.9103	2.5131
13	Logsig	2	0.8142	68	0.9584	0.8237	0.8740	4.2739	0.8135	3.1225	0.8150	4.1104
14		3	0.8468	58	0.9197	0.8994	0.8915	3.0264	0.8413	2.4166	0.8564	3.9061
15		4	0.8762	48	0.9328	0.5449	0.9127	2.4617	0.8734	1.9124	0.8643	3.3415
16		5	0.9010	44	0.9385	0.5140	0.9338	2.1232	0.8980	1.6816	0.8876	3.0413
17		6	0.9170	43	0.9478	0.4458	0.9457	2.0833	0.9198	1.6132	0.9120	3.0922
18		7	0.9271	36	0.9539	0.4363	0.9600	1.8385	0.9384	1.4448	0.9320	2.9298
19		8	0.9338	38	0.9582	0.3943	0.9620	2.0749	0.9411	1.4663	0.9359	2.8583
20		9	0.9363	36	0.9620	0.3795	0.9682	1.6762	0.9519	1.2627	0.9409	2.6781
21		10	0.9412	**30**	0.9650	0.4118	0.9720	1.7408	0.9604	1.3396	0.9468	2.7705
22		11	0.9459	34	0.9665	**0.3551**	0.9760	1.6523	0.9651	1.2436	0.9490	2.7766
23		12	**0.9500**	33	**0.9682**	0.3631	**0.9786**	**1.6521**	**0.9695**	1.1869	**0.9555**	2.5246
24		13	0.9432	35	0.9641	0.3728	0.9731	1.7239	0.9543	1.2165	0.9461	2.6987
Types of activation functions			$\mathrm{Tansig}(x)=\frac{2}{1+\exp(-2x)}$ $\mathrm{logsig}(x)=\frac{1}{1+\exp(-x)}$									

**Table 4 foods-12-01897-t004:** Comparison results between BP model and GA-ANN model.

Algorithm	DT		SEC		HR		WR		SR	
	*R* ^2^	*RMSE*	*R* ^2^	*RMSE*	*R* ^2^	*RMSE*	*R* ^2^	*RMSE*	*R* ^2^	*RMSE*
BP	0.950	33	0.968	0.363	0.979	1.652	0.970	1.187	0.956	2.525
GA-ANN	0.996	22.176	0.998	0.279	0.990	1.471	0.991	0.926	0.993	1.250
Error (%)		32.8		23.14		10.96		21.99		50.5

**Table 5 foods-12-01897-t005:** Pareto optimal set (shaded groups are the optimum sets in the Pareto front).

Pareto ID	T (°C)	V (m/s)	MC	D	DT (min)	SEC (MJ/kg)	HR	WR	SR
1	51.5	1.04	10.0	2	958.4	16.77	89.0	50.9	92.8
2	53.3	1.20	10.1	2	888.5	16.49	88.9	50.4	93.6
3	47.0	1.01	10.0	2	1082.2	16.78	88.8	51.2	89.5
4	44.7	1.50	11.1	2	974.0	14.77	86.2	48.4	86.8
5	52.5	3.59	23.2	2	137.9	1.84	48.8	24.9	68.2
6	43.1	1.02	10.0	2	1193.4	16.78	88.7	51.2	85.3
7	54.9	3.66	10.9	3	585.1	12.86	81.7	43.7	99.8
8	54.9	3.98	20.3	3	181.4	2.78	58.6	32.1	84.8
9	54.7	3.97	21.3	2	157.2	2.33	54.1	28.8	75.7
10	49.6	1.01	10.0	2	1012.0	16.79	88.9	51.0	91.6
11	53.4	3.83	17.4	2	264.8	4.70	69.2	38.3	85.3
12	54.7	3.85	23.9	2	109.6	1.29	47.8	23.8	66.1
13	54.2	3.25	25.0	3	103.9	1.08	56.8	29.3	69.3
14	52.5	2.27	16.0	2	407.9	7.45	75.9	41.6	87.3
15	54.9	3.13	17.0	2	299.7	5.65	72.3	39.5	87.0
16	50.6	2.16	10.2	2	847.7	15.48	87.5	49.1	92.9
17	47.7	1.97	12.8	2	696.3	11.68	82.5	45.4	88.3
18	54.9	3.82	24.2	2	102.8	1.15	47.3	23.3	64.6
19	52.5	2.92	12.9	2	535.3	10.54	82.1	44.9	91.9
20	43.6	1.39	14.2	2	729.8	10.64	80.2	44.0	82.1
21	49.8	3.38	12.4	2	573.7	10.85	82.0	44.6	91.3
22	43.6	1.02	10.1	2	1169.3	16.65	88.6	51.0	85.8
23	47.3	1.75	10.9	2	901.0	14.80	86.4	48.5	89.7
24	50.4	3.71	17.2	2	299.8	5.11	69.5	38.4	84.0
25	52.9	3.65	20.4	3	200.6	3.06	58.6	32.2	83.5
26	52.5	2.74	16.6	2	353.5	6.39	73.3	40.3	86.3
27	52.7	2.03	15.7	2	436.6	8.03	77.2	42.3	87.9
28	55.0	3.98	23.3	2	116.8	1.45	48.7	24.7	68.5
29	54.9	3.93	23.0	2	123.3	1.60	49.4	25.2	69.6
30	55.0	3.99	25.0	2	87.3	0.81	46.1	22.4	61.8

**Table 6 foods-12-01897-t006:** Prediction and validation results of five response variables under the optimal parameters.

Results	Operating Conditions				Response Variables				
	T (°C)	V (m/s)	MC	D	DT (min)	SEC (MJ/kg)	HR	WR	SR
Prediction	54.9	3.66	10.9	3	585.1	12.86	81.7	43.7	99.8
validation	54.9	3.66	10.9	3	565	12.38	83.8	45.2	96
Error (%)					3.56	3.88	2.51	3.32	3.96

## Data Availability

The data presented in this study are available on request from the corresponding author.

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
