# Peer review of "Artificial Neural Network Modeling and Genetic Algorithm Multiobjective Optimization of Process of Drying-Assisted Walnut Breaking"

_foods, 2023, doi:10.3390/foods12091897_

Round 1
Reviewer 1 Report
The article propose a modeling and optimisation methodology for walnut breaking optimisation that is sound and original. The experimental works seems well realized and provides data of a quality high enough for the training of the model. If the exact optimization presented in the paper is of interestfor some readers, it is more the followed methodology that is of interest for a larger audience.
Yet, the manuscript has a several problems in its presentation and organization. Therefore I strongly recommentd to enhance drastically the text. Particularly, the Material and Method section requires a lot more precisions. Here are first a list of general improvement
- the title is not easy to understand, a title like "Drying conditions impact on walnut breaking: ... " is probably more understandable.
- in general, in the text, it should be sufficient to say once that drying is a pretreatment and then just use the wrod drying, instead of drying pretreatment assisted ...
- introduction should explain why a neural network is needed and a simplier model, or a physic based model is not sufficient.
- equations 16 to 24 are not at their place in the result section, they should be introduced and explained in Material and Methods
- in the results, all the numerical values that are allready in the tables should not be repeated in the text. Just point out the optimal conditions without repeating all their characteristics.
- you give physical explaination to the variations that are shown in figure 5 to 7 but they are just litterature based assumptions youmake, you don't have any verification. This should be made clear in the way the explainations are given
- Tables with the weights should be in appendix, they are not discussed in the text and are of little interest for a general reader
- it is written that all the used data are in the paper, which does not seem to be correct. we don't have the full 192 groups of results used to train the model.
- some sentence are not understandable. Please have a systematic correction by a native speaker to make it fully understandable.
On top of this general comments, I have the following detailed points that should be adressed :
- in the introduction, many concepts are not properly defined : shell-breaking rate, high kernel rate, whole kernel rate and for the ANN : weights and thresholds. They should be at least qualitatively explained or you should not use them before introducing them in the Material and Methods
- what IR temperature represents practically is not clear. If it is just a machine parameter, it is insufficient for another team to reproduce the results. Is it the air temperature ? or the product surface temperature ? this should be precised and explained.
- nozzle air velocity is not sufficient to characterize the air flow : air flow rate and internal part cross sections should be precised
- When speaking about moisture content, most of the time, it is the final moisture content that is given (10%, 15M, 20% or 25%), but this is fully implicit in the paper.
- the definition of Dt from equation (7) is not precise enough and will not allow reproduction of the test by another team, please be more complete.
- The way equations (8) to (11) are used is unclear, as they are contradictory. Please explain.
- 192 groups of data are used, but what are these data, how are they classified, grouped should be precised
- in figure 6, y-axis left legend should be High Kernel Rate, not High exposure rate
- Material and methods explains only your approach of training by GA-BP while results compare BP and GA-BP. Material and method should be updated accordingly
- as pureline just equals x, equation 18 and 19 should be combined and simplified in one simple equation (that should appear in material and method)
- You have all along the paper variables that are in capital letters sometime but not all the time (t-T, x-X), please make that uniform
Reviewer 2 Report
“Drying pretreatment assisted walnut breakings: artificial neural network modeling and genetic algorithm multi-objective optimization” was the goal of this work. The manuscript has to be changed as follows:
· The drafting of the abstract did not follow the paper's formatting instructions. This section should contain a summary of the investigation's findings, both qualitatively and quantitatively.
· The introduction is not sufficiently clear. This section should include a description of any relevant earlier work. , add Multivariate optimization of mechanical and microstructural properties of welded joints by FSW method, 10.1533/9780857094551.543, Artificial neural networks based prediction of performance and exhaust emissions in direct injection engine using castor oil biodiesel-diesel blends
· In the introduction part, include recent literature released after 2018
· This article needs to be edited with attention.
· The quality of the figures must be improved.
· The test conditions must be fully described.
· By applying this existing methodology, the conclusion needs to be written more precisely
· Results and discussions must be supported by references to accepted literature.
· What criteria are used to select input parameter ranges? References must be provided if the input parameters were based on earlier studies.
· The results weren't sufficiently discussed. Each part must contain a suitable statement of the findings..
· This field has been extensively researched. How does this research contribute to innovation? In the introduction, this innovation should be explicitly stated.
· The findings must be strongly supported by evidence from the literature and inference.
Round 2
Reviewer 2 Report
The manuscript is acceptable.